# Acoustophoretic system for seed separation on conveyor belts

**James Hardwick** [1,2] ✉, **Zak Morgan** [1,2] & **Ryuji Hirayama** [1,2]

Seed sorting on conveyor belts is a vital part of the modern agriculture industry, allowing for substandard seeds to be identified and ejected before germination. Conventional sorting systems typically rely on air-jet ejection, which can be expensive and limited in both accuracy and flexibility. Ultrasonic phased array transducers offer an alternative solution for seed separation on conveyor belts that addresses many of these issues. These arrays can generate focused beams of sound energy that affect small objects through acoustic radiation forces and offer precise, programmable control. Here we show that by combining phased arrays with a camera detector, seeds can be selectively identified and acoustically ejected based on quality metrics such as size and color. Unlike binary sorting methods, our system allows for variable ejection distances, enabling more nuanced categorization. We validate our approach through extensive experiments in both static and dynamic conditions, demonstrating effective separation not only of sub-standard seeds but also of unwanted debris. This acoustophoretic system is low-cost, adaptable, and capable of advanced functionalities such as mid-air hovering for seed alignment, full levitation for quality inspection or coating, and ejection in a free-fall environment. Our findings suggest a promising direction for next-generation agricultural automation and precision sorting systems.

Sorting on conveyor belts is a fundamental process across various industries, where materials or products are separated based on specific criteria as they move along a conveyor line. This method is widely used in manufacturing[1], mining[2,3], recycling[4–6], and agriculture[7,8] to categorize objects, typically ensuring that only items meeting specific quality standards move on to the next production stage. In the agricultural sector, sorting is particularly vital for seed selection, where it plays a crucial role in ensuring that only high-quality seeds are chosen for planting[9,10]. Seed sorting ensures that substandard seeds, which may affect germination and crop health, are identified and removed before they reach the soil. This process is essential for maximizing crop yield, maintaining plant health, and optimizing resource use, making it a cornerstone of modern agricultural practices and the subject of much recent research[11–13].

Seed sorting is crucial for modern agriculture, but current systems face challenges to efficiency and cost-effectiveness, particularly with small, delicate seeds. Before the advent of modern high-speed sensors and detection software, gravity sorters were the state-of-the-art for small object sorting. These systems use vibration to separate items based on mass, shape, and density, with denser objects migrating to different areas of a vibrating plate. While still a low-cost (~ $10–30k) option for sorting low-value seeds or grains, gravity sorters lack the precision needed to distinguish objects with similar densities. For products where high germination rates and seed quality are essential (such as tomato seeds, for example), this level of inaccuracy is unacceptable, making optical sorters the preferred choice for high-precision applications.

Typical optical sorting machines combine sensors, such as RGB color, near-infrared, hyperspectral, and laser imaging, with pneumatic jets that separate defective seeds based on detected features like size, color, and shape[14]. Pneumatic separators consist of an array of nozzles with an aperture and pitch on the order of a few mm. Each nozzle has a

[1]Department of Computer Science, University College London, London, UK. [2]Acoustofab Ltd, London, UK. ✉e-mail: james.hardwick.19@ucl.ac.uk

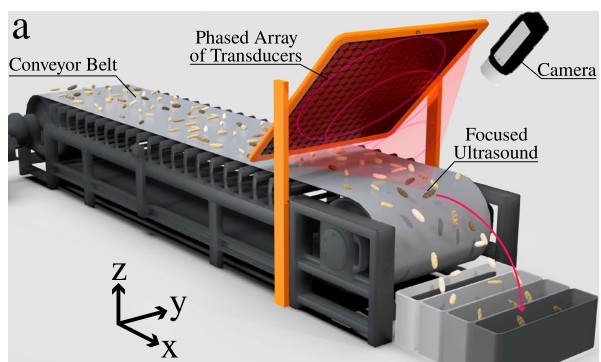
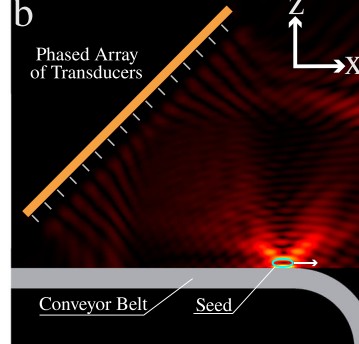

**Fig. 1 | Acoustophoretic separation system. a** Schematic concept of our separation system using focused ultrasound wave to selectively eject seeds. **b** Simulation of the sound pressure distribution when ejecting a seed.

cost in the region of $100, and 50–100 nozzles are typically needed to cover the full width of a conveyor belt. As such, with the added cost of a compressor to feed them with compressed air (~ $1000), a full air-jet separation system will cost in the region of $6–11k. While highly effective, even when sorting objects of similar size, shape or mass, day-to-day operation of an air-jet separator is costly due to susceptibility to clogging from dust and particulates, intricate moving parts with frequent maintenance needs, and the storage and deployment of compressed air[15,16]. The limited aperture and pitch of air-jet nozzles can also reduce ejection effectiveness, as high quality seeds can often be ejected along with low quality seeds or detritus during the separation process.

Recent research has sought to address some of the challenges posed by pneumatic jets. Robotic pick-and-place systems[17,18] have improved their precision in sorting tasks through technologies like simPLE[19] for multi-observation of sorting objects, and decision-making in uncertain environments, such as randomly distributed objects on the surface of a conveyor belt[20]. Yet, these systems remain relatively slow, handling one object at a time, and are costly, often exceeding $100,000, making them inefficient and impractical for high-speed, delicate separation tasks. For separation of delicate objects such as seeds, recent developments in soft robotic actuators[21] and end-effectors[22] have been promising. This is a cheap and scalable way to enable the separation of small, fragile, or irregularly shaped objects, without causing damage to them.

In 2023, Zournatzis et al. introduced a soft robotic actuator for integration with conveyor belts[23]. Their SoftER is a soft pneumatic actuator designed to rapidly unwind when pressurized, delivering quick and powerful impact forces. Its design, inspired by natural spiral motions, offers a simple, low-cost alternative to pneumatic actuation systems. However, while robotic pick-and-place systems are precise, they are slow and expensive, making them inefficient for high-speed sorting of small, delicate items like seeds. Similarly, soft robotic actuators, though gentler and more cost-effective, may still struggle to meet the high-speed and high-precision demands for efficient sorting of lightweight seeds on conveyor belts.

This study introduces an approach to seed separation that leverages ultrasonic phased array transducers (PATs)[24] as the primary component in an acoustic sorting system. Figure 1a illustrates our system, where a PAT focuses ultrasound waves onto seeds at the edge of a conveyor belt, altering their trajectories to sort them into multiple bins. PATs are a powerful tool in various applications, known for their ability to manipulate objects with precision using sound waves, as well as offering precise control of ultrasonic wavefronts over a certain operating volume (see Fig. 1b). These systems consist of large arrays of small, individually controlled speakers that can manipulate the phase and amplitude of sound waves, creating intricate pressure landscapes through constructive and destructive interference[25,26]. In our

acoustophoretic separation system, PATs operate outside the range of human hearing in the ultrasonic regime (in our case 40 kHz) to generate beams of acoustic radiation which impart force to selectively eject small objects like seeds. They offer high-speed operation and remarkable precision, making them an excellent choice for applications requiring delicate and accurate handling of small objects.

Acoustic waves are already widely used in acoustofluidics for cell and particle separation due to their contactless, label-free features[27]. In these applications, sorted objects are typically in the micrometer scale, using frequencies of several MHz within a liquid medium. Our approach, however, focuses on millimeter-scale objects in air (e.g., on a conveyor belt) using 40 kHz PATs. This introduces unique challenges due to the increased size of the objects, such as the greater forces required to eject them. There has been extensive prior work using similar PAT systems at 40 kHz for acoustic levitation in mid-air[28–30] across various applications, including displays[24–26], liquid handling[31], and 3D printing[32]. There are a few studies using PATs for pushing centimeter-scale objects in the field of human-computer interaction (e.g., polystyrene balls[33], ping pong balls[34], and balloons[35]).

By integrating PATs as the separation component in the optical sorting process, we aim to enhance efficiency, accuracy, and cost-effectiveness, especially compared to state-of-the-art air-jet approaches, which are their most similar counterpart. Our approach not only simplifies the separation mechanism by reducing reliance on mechanical parts, but also offers greater flexibility in handling seeds of different sizes, shapes, and conditions. By utilizing ultrasonic waves to selectively eject substandard seeds from the conveyor belt, our method promises to overcome the speed and accuracy limitations of traditional systems, offering a significant improvement in sorting performance.

The core of our contribution lies in demonstrating how PATs can be used to selectively eject bad seeds from a moving conveyor belt, independently of the specific sensor detection system used. While a camera detector identifies defective seeds based on visual cues in our current sorting setup, our ejection system is designed to be flexible and can function alongside any current or future sensor detection system. Our primary innovation lies in precisely targeting and ejecting seeds using focused sound waves from PATs, eliminating the need for complex mechanical or pneumatic systems, reducing costs and enhancing reliability.

To validate the effectiveness and adaptability of our acoustophoretic sorting system, we conducted extensive experiments across a range of scenarios. Static experiments demonstrated precise spatial resolution for our PATs, displaying selective ejection of seeds within densely packed arrangements. Dynamic tests evaluated the ability of a full acoustophoretic sorting system to sort moving seeds by manipulating their trajectories based on weight and size, achieving non-binary sorting and detritus removal with accuracy, purity and yield

ratios of 68–100% in all cases. These experiments evaluated the system's ability to sort seeds of different sizes and conditions under various operational environments. The results show that ultrasonic phased arrays provide a robust, versatile solution with the potential to transform seed sorting, particularly by addressing limitations of pneumatic air jet separators, which also rely on precise actuation for separation. In the future, capabilities like hovering for pre-arrangement and 3D levitation for inspection and coating could enhance seed processing efficiency, reduce mechanical complexity, and enable multifunctional agricultural applications.

The introduction of our acoustophoretic seed separation system has significant implications for the future of agricultural technology. By providing a more efficient and accurate method for seed separation, this innovation can lead to higher crop yields and better resource management. Moreover, the system's ability to perform advanced operations like 3D scanning and precise seed coating opens the door to new applications in seed treatment and quality assurance. As agriculture continues to evolve in response to global challenges, such as food security and sustainability, innovations like our acoustophoretic separation system will play a crucial role in driving progress. This technology not only addresses current limitations but also creates opportunities for further advancements in seed processing, making it a valuable addition to the toolkit of modern agriculture.

## Results
To evaluate the performance of our acoustophoretic sorting and seed ejection system, we undertook a series of experiments, in which we used different types of seeds. The average size (surface area), thickness, and weight of each seed type measured are summarized in (Table 1) as follows:

**Table 1 | Summary of seed and other sorting object properties**

| Type | Cross-section [mm²] | Thickness [mm] | Weight [mg] |
|---|---|---|---|
| Cabbage | ~2.6 | 1.58 | 3.5 |
| Melon | ~38 | 1.91 | 49.2 |
| Cucumber | ~27 | 1.51 | 28.2 |
| Tomato | ~7 | 0.82 | 3.3 |
| Pepper | ~11 | 0.82 | 5.9 |
| Coriander | ~9–13 | 2.72–3.29 | 12.6 |
| Stone | ~20–36 | 2.22–3.61 | 56–168 |
| Stick | ~3–8 | 0.41–0.54 | <0.1 |

In the cells of this table, single values represent means, and two values represent a range between minimum and maximum when individual items of classes are too dissimilar to be easily compared.

Throughout our experiments, the size of the generated focal point significantly influenced the selective ejection of seeds. For example, when creating a focal point 12 cm from the center of the array (see Fig. 2a), the values of full width at half maximum (FWHM) in the horizontal and depth directions were 0.9 cm and 4.7 cm, respectively. Although the maximum pressure depended on the distance from the array (peaked around 9 cm as shown in Fig. 2b), the phased array system can deliver high pressure amplitude even at large distances. In addition, the PAT can create a high-pressure focal point (>5 kPa with a distance of 12 cm) even at the edge of the board ($Y = \pm 8$ cm), as shown in Fig. 2c. By modulating pressure amplitudes, the system can precisely control the strength of the acoustic radiation force applied to seeds, determining both the distance and velocity of ejection. These factors underline the system's capacity for flexible, non-contact manipulation, and enable its application to each of the experiments and use-cases we present in the following sections.

### Static ejection experiments
In our first set of experiments, we used a static setup, as shown in Fig. 3a to examine the horizontal and depth resolution of ejection. Here, a PAT board was tilted 45 degrees, facing a flat surface with a grid. The distance between the board and the grid center was 12 cm. At the center of the grid, three pepper seeds were arranged horizontally (i.e., along the $Y$-axis) at equal intervals. We refer to the distance between the centers of adjacent seeds as $\Delta Y$. We chose to use pepper seeds for this experiment because of their relatively uniform, rounded shapes, which make it easier to measure $\Delta Y$. A focal point was then generated at the location of the central seed for 20 ms, which is slightly longer than the ejection time used to eject all seed types in a later experiment (detailed in Section "On-belt ejection experiments") to ensure enough time to eject the central one. The ejection is considered successful only if the central seed alone can be ejected by more than 3 cm without affecting the other two seeds (i.e., displacing them by more than 5 mm). $\Delta Y$ was gradually reduced from 10 mm in increments of 1 mm, and the minimum value of $\Delta Y$, at which the test is successfully completed three out of three trials was defined as the horizontal resolution.

As a result, the experiment was successful in all three trials when $\Delta Y$ is 7 mm or more, but only in two out of three trials when $\Delta Y = 6$ mm. Therefore, the minimal distance between seeds allowed for selective

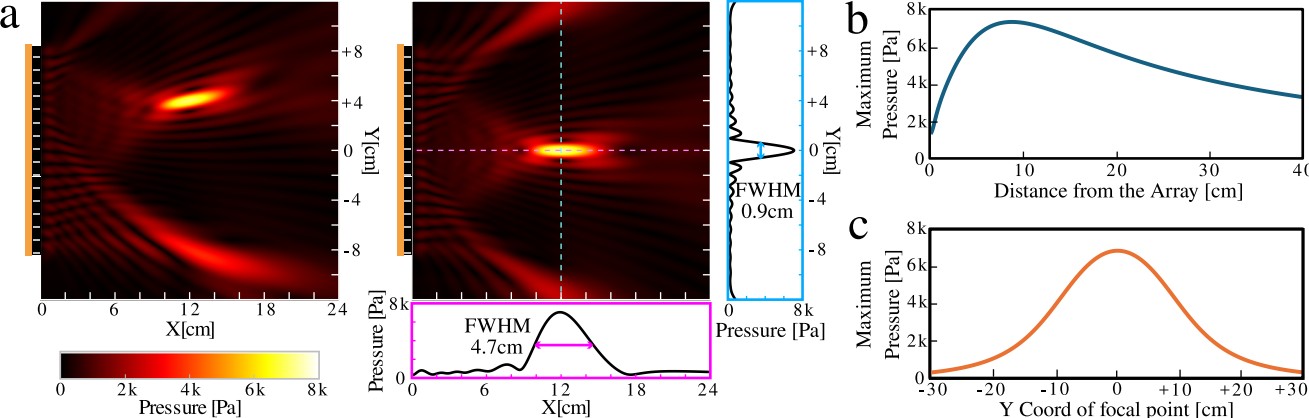

**Fig. 2 | Acoustic characteristics of the phased array transducers (PAT). a** Sound-field simulations when creating a focal point at $Y = +4$ cm (left) and $Y = 0$ cm (right) using the transducer piston model. The pressure amplitude values along the horizontal (Y) and the depth (X) axes, as well as their full width at half maximum (FWHM) values, are presented. **b** Acoustic pressure amplitude of a focal point at different distances from the PAT. **c** Acoustic pressure amplitude of a focal point at different Y-coordinates. Source data for this figure are provided in Supplementary Data 1.

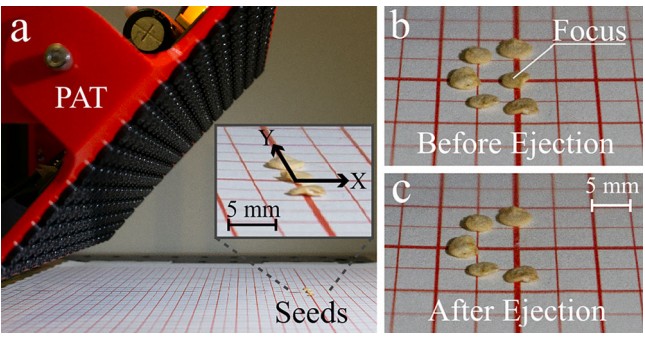

**Fig. 3 | Static experiments. a** Experiment setup. **b** Close-up photos of densely arranged seeds before and **c** after seed ejection, demonstrating selective ejection even with a packed arrangement.

ejection was defined as 7 mm. This value is close to the horizontal FWHM value (= 9 mm) shown in Fig. 2a.

Next, to investigate the depth resolution of ejection, we conducted a similar experiment with seeds arranged along the *X*-axis. As shown in Fig. 2a, the acoustic focal point is elongated in the direction perpendicular to the PAT (i.e., along the ultrasound propagation axis), with the FWHM approximately five times larger than in the horizontal direction. If this propagation direction were aligned with the *X*-axis, the minimum distance between seeds in the depth direction would be much larger than in the horizontal one. In our intended setup (Fig. 1), however, the PAT is tilted so that seeds located closer to the PAT than the focal point (in the negative X direction) are less affected by this elongated focal point. Although seeds positioned farther from the focal point would still be affected by it, the focal point is created at the edge of the conveyor belt, so such cases do not need to be considered. Thus, instead of using three seeds, we used two pepper seeds in this experiment and obtained $\Delta X = 5$ mm.

Figure 3b, c demonstrates that the PAT board can eject individual seeds with minimal disturbance to adjacent ones, even at centroid-to-centroid spacings as small as 7 mm laterally and 5 mm longitudinally. This level of spatial precision is comparable to a tightly packed monolayer of seeds, with little or no gap between them. Such spacing metrics are only relevant to systems that actuate objects individually from a monolayer—most notably pneumatic air-jet sorters, which similarly rely on targeted ejection. In contrast, gravity sorters operate via bulk flow on inclined vibrating planes and sort based on mass, shape, or density rather than precise per-object control. Because objects in gravity sorters are typically packed in multiple layers and in constant contact, spatial resolution is not a meaningful metric in that context. For monolayer sorting applications, our system achieves a resolution comparable to pneumatic systems, with the added benefit of precise, selective actuation and minimal collateral displacement. In future designs, increasing the PAT device's operating frequency, and thereby decreasing its wavelength, could further narrow the ejection beams, enhancing separation accuracy.

**On-belt ejection experiments**

In the next set of experiments, we tested the ejection potential of the PAT board on seeds moving along a conveyor belt. The experimental setup for these experiments is shown in Fig. 4a and featured a single PAT board mounted above a conveyor belt. The PAT altered the seeds' falling trajectories by directing pulses of focused ultrasound at specific targets. The camera positioned above the board, combined with object tracking software, monitors seed positions on the conveyor belt and fed this data to the computer controlling the field generated by the PAT board. The conveyor belt was positioned 33 cm above the table. We oriented the board so that a 12 cm line could be drawn from the center of the conveyor belt's edge along the *Y*-axis (orange dot in

Fig. 4a) to the board's midpoint. Seeds actuated by the PAT experienced an acoustic force that altered their natural trajectory, causing them to fall further to the right. In contrast, unactuated seeds followed their natural trajectories, falling further to the left. We set out to evaluate the board's performance in ejecting a variety of objects (shown in Fig. 4b). From left to right, the materials include three representative examples each of pepper seeds, tomato seeds, coriander seeds, sticks, stones, and glass shards. The non-seed objects are included as common forms of detritus that can be mixed in with populations of seeds in a factory environment.

In order to test the robustness of the system to latency, a single capsicum pepper seed was placed on the end of a static conveyor belt. Placement on the raised conveyor rather than a flat surface, increases the time the particle spends falling, thus increasing the differentiation in change of horizontal ejection distances. Focal points were then created for 10 ms with the maximum ejection pressure (~ 7 kPa), with varying offsets to the object's center in the direction of the ejection axis (*X*-axis). If we have poor latency in our system, our focal points will lag behind the particle to be ejected, and if we over compensate, it will track in front of the particle. We varied the offset from −10 to +10 mm, approximately the size of the FWHM value of the focal point, to test how location of the focal point against the seed affects ejection distance.

As shown in Fig. 4c, it was found that when the offset was in front of the center of mass of the particle by 2 mm or more, ejection failed to occur. This is due to the focal point passing beyond the particle's edge. When the focal point is offset increasingly behind the particle's center, the ejection distance initially rises until the focal point aligns with the particle's edge. After this, ejection distance decreases almost linearly, finally failing to eject when 10 mm in front of the particle. Again, this is expected due to the size of the focal point. Error bars in this plot represent the spread of ejection distances recorded over 5 trials for each offset value.

Knowing these distances, we find that the system's sensitivity to latency depends on conveyor speed. At the slowest speed of 1 cm/s used in our experiments, an assumed 100 ms latency would result in a position error of about 1 mm, causing a 20 mm (16.7%) reduction in ejection distance. Whilst this value is not small, it is not thought to have impacted our experiments greatly. At the fastest conveyor speed of 15 cm/s, a 100 ms latency would likely result in frequent ejection failures, emphasizing the importance of maintaining low and consistent latency for high-throughput systems.

In order to characterize the ability of our system to eject these objects by controllable distances, we set up two experiments, in which two parameters are varied. In the first part of the experiment, we explored the effect of varying the pulse duration of the PAT, that is, the time window (in milliseconds) that the PAT board imparted acoustic pressure onto an object. In the second part, we explored the effect of varying the pressure amplitude fed to the transducers, which directly influences the device's overall pressure output. Both of these parameters represent ways to control the intensity of the focused sound waves over time and thus the acoustic force imparted on an object by the PAT. In both experiments, individual objects were sent down the conveyor belt, consistently positioned centrally along the *Y*-axis. Ejection occurred at the end of the conveyor (the maximum *X*-position) just before the object began its natural descent. In the pulse duration experiment, the transducer pressure amplitude was fixed at its maximum value of 7 kPa, with results for all six object types shown in Fig. 4d. In the amplitude experiment, the pulse duration was held constant at 15 ms, and the corresponding results are presented in Fig. 4e. The displacements of each object were measured by placing a sticky board beneath the outlet of the conveyor. The board, made of upward-facing adhesive strips, captured falling objects at their landing positions, preventing bouncing and enabling precise measurement of their X-coordinates. The recorded displacements represent the

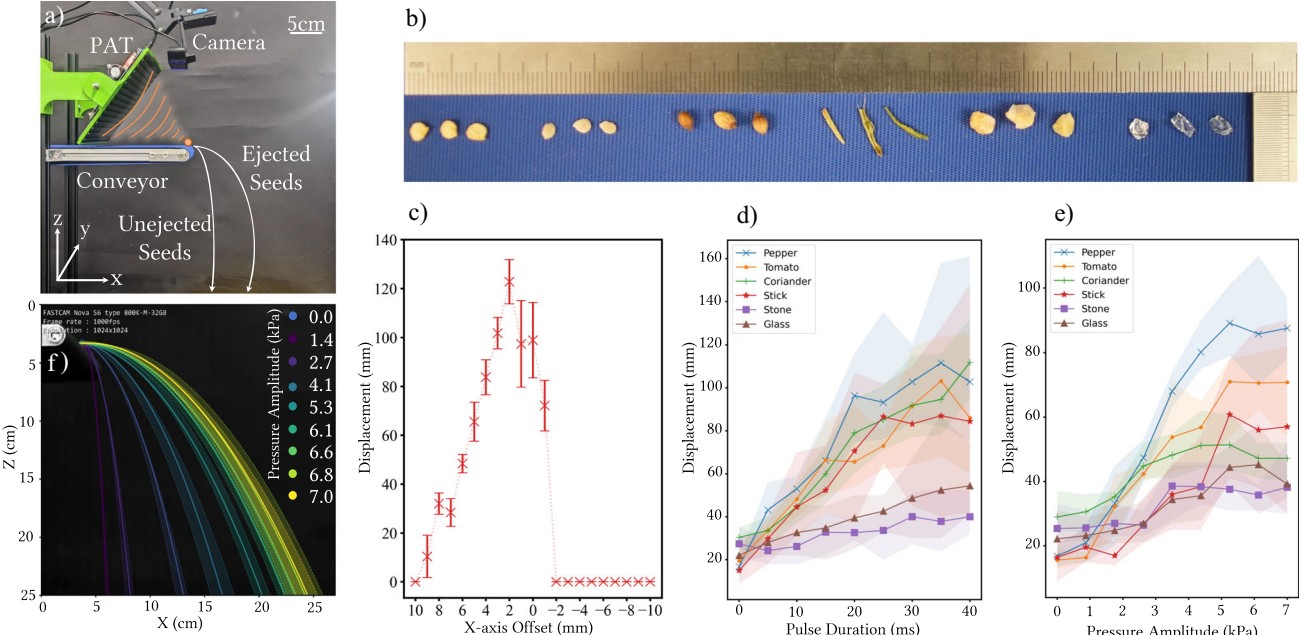

**Fig. 4 | On-belt pulse and amplitude experiments. a** Experimental setup for these experiments with components labeled. **b** Typical sizes of various seeds and other objects described in this report. From left to right in groups of three, pepper (capsicum), tomato, coriander, stick, stone, and glass shards. **c** Effect of varying the positional x-axis offset of the focused ultrasound from the phased array transducers (PAT) on ejection displacement of a capsicum seed. The reported value is the average of five trials, with error bars indicating the range of displacement measurements across those trials. **d** Effect of varying the pulse duration and **e** pressure amplitude of the PAT on the ejection displacement of different materials. **f** A capsicum pepper seed is ejected with varying pressure imparted by the PAT board, from 0 Pa to 7 kPa at the focal point. Source data for this figure are provided in Supplementary Data 2.

X-dimension distances beyond the conveyor belt's final X-coordinate. For each object, five trials were conducted at each pulse duration in the first experiment and each transducer amplitude in the second. The solid lines in Fig. 4d, e indicate the average distances across these trials, while the shaded areas show the ranges.

For pulse duration (Fig. 4d), the displacement increase is approximately linear, with smaller, lighter objects such as seeds and sticks showing a greater response compared to heavier stones and glass shards. However, this increase plateaus at higher pulse durations (30–40 ms), likely because objects are ejected and pushed away from the area of focused acoustic pressure within ~30 ms, rendering the remaining pulse duration redundant. For amplitude (Fig. 4e), low pressures (1–2 kPa) produce negligible effects, as they fail to generate sufficient force to move objects. As amplitude increases to moderate levels (3–4 kPa), displacement rises significantly, particularly for lighter objects. Beyond 5 kPa, the displacements plateau probably due to the PAT system's non-linear pressure behavior[36].

In a third experiment, we wanted to better visualize the effect of varying amplitude on one of these objects. We chose capsicum pepper seeds as our ejection object, as these are relatively flat, round, high-value seeds that a product developed from our prototype system may one day be used to sort. As previously, the ejection position was set at the edge of the conveyor belt. The pulse duration was fixed at 10 ms. Seeds were then ejected, one by one, from the central position (in the *Y*-axis) of the belt, with variable pressure amplitude values. In total, 45 ejection experiments were undertaken, with 5 trials for 9 different pressures, ranging from 0 Pa (no ultrasound emanating from the board) to the maximum pressure of 7.0 kPa. We used 0, 1.4, 2.7, 4.1, 5.3, 6.1, 6.6, 6.8, and 7.0 kPa due to the system's non-linear response. Crucially, unlike the previous experiment (Fig. 4e) we filmed the trajectories of the ejected seeds using a high-speed camera and used a custom OpenCV-based tracking software to record their trajectories. The recorded trajectories can be seen plotted in Fig. 4f. As expected,

higher pressure values result in greater ejection distances, with a relatively steady increase.

These three experiments show that lighter objects like seeds and sticks exhibit greater displacement with increasing pulse duration or amplitude compared to heavier objects. However, the effects plateau at higher pulse durations (>30 ms) and amplitudes (>6 kPa) due to pulse redundancy and system non-linearities. High-speed camera recordings of seed trajectories support these findings and highlight the system's potential for non-binary sorting. Objects from different categories could potentially be directed to three, four, or even five different bins with consistent accuracy.

## Detritus extraction

One potential application for our system is in the removal of detritus, such as sticks and stones, from seed populations. Traditional methods, especially air-jet systems, face limitations to their separation accuracy such as cost, noise, and inaccuracy[14]. To address these challenges, we performed a detritus extraction experiment, separating a population of seeds from unwanted sticks and stones. As shown in Fig. 5a, a mixture of coriander seeds, sticks, and stones was placed on a 12 cm-wide conveyor belt. Object detection measured sphericity, assigning values between 0 (non-spherical) and 1 (perfectly spherical). Sticks, with low sphericity, fell into the reject bin, while seeds and stones, exceeding the threshold of 0.5, were subjected to acoustic forces. The force imparted by the beam was calibrated to be ineffectual against heavy stones, which continue to fall with their natural trajectories into the reject bin, while lighter seeds are affected by the focussed sound and are pushed into the accept bin.

In this case we assessed sorting performance using two standard metrics in sorting literature: purity and yield[37]. TP denotes the number of true positives (i.e., desirable seeds correctly accepted), FP the number of false positives (i.e., undesired items incorrectly accepted), and FN the number of false negatives (i.e., desirable seeds incorrectly rejected). We define Purity as the proportion of correctly accepted

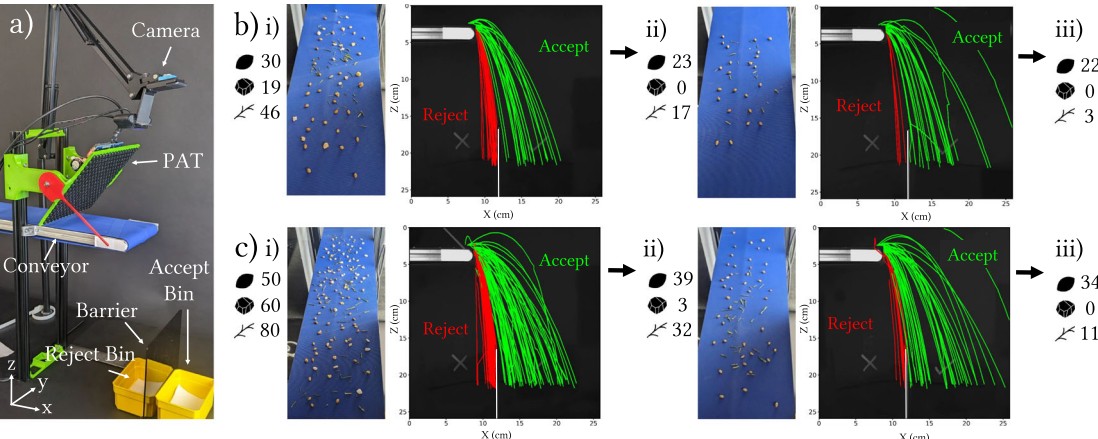

**Fig. 5 | Detritus extraction experiments. a** Experiment setup. Seeds are sent to the accept bin to the right of the barrier. Stones and sticks are sent to the reject bin to the left. **b** (i) In the first experiment 30 seeds, 19 stones and 46 sticks are loaded onto the conveyor belt and passed through the ejection system. Of these, 23 seeds and 17 sticks are retained in the accept bin and the rest of the objects are rejected. (ii) The contents of the accept bin are then passed back through the ejection system. (iii) Of these, 22 seeds and 3 sticks are retained. This experiment results in an 88% purity and a 73% yield. **c** (i) In the second experiment 50 seeds, 60 stones and 80 sticks are loaded onto the conveyor belt and passed through the ejection system. Of these, 39 seeds, 3 stones, and 32 sticks are retained in the accept bin and the rest of the objects are rejected. (ii) The contents of the accept bin are then passed back through the ejection system. (iii) Of these, 34 seeds and 11 sticks are retained. This experiment results in a 75% purity and a 68% yield.

desirable seeds among all accepted items:

$$\text{Purity} = \frac{TP}{TP + FP} \qquad (1)$$

and yield as the proportion of desirable seeds that are successfully accepted:

$$\text{Yield} = \frac{TP}{TP + FN} \qquad (2)$$

These metrics provide an assessment of system performance in line with how state-of-the-art sorting systems are typically evaluated.

We conducted two experiments, each with two passes through the acoustophoretic sorting system, to evaluate its effectiveness. In the first experiment (Fig. 5b), we used a mixture of 30 coriander seeds, 19 stones, and 46 sticks (initial purity of 31%). After the first pass (i), the accept bin contained 23 seeds, 0 stones, and 17 sticks. After a second pass (ii), the final count was 22 seeds, 0 stones and 3 sticks, resulting in an 88% purity and a 73% yield. In the second experiment (Fig. 5c), we increased the mixture to 50 seeds, 60 stones, and 80 sticks (initial purity of 29%). After the first pass (i), 39 seeds, 3 stones, and 32 sticks were accepted. A second pass (ii) resulted in 34 seeds, 0 stones, and 11 sticks, giving a 75% purity and a 68% yield.

We intentionally chose unrealistically high ratios of detritus in the mixtures we tested. A typical ratio of detritus in a factory environment would be more like 5–15%[38]. We chose the exaggerated ratios to stress-test the system's performance under more challenging conditions. Despite high levels of detritus, the acoustophoretic sorting system performed well, achieving up to 73% and 68% yields in the first and second experiments respectively and more than doubling purity in both experiments after two passes.

Several factors likely contributed to the system not achieving even greater performance. One likely cause is that some objects stick to the surface of the conveyor belt, counteracting the acoustic force and causing them to fall short of their target when they are ejected. Furthermore, very densely packed objects can sometimes be ejected along with neighbors, as we show in Section "Static ejection experiments". This can cause sticks, for example, to occasionally be ejected along with seeds and end up in the accept bin, or seeds to fall short of

after colliding with other seeds in the air. Finally the camera used for optical inspection ran at a low framerate of 30 Hz or below depending on lighting conditions, and the latency of the system was variable, resulting in non-uniform ejection distances for similar objects.

Despite these factors, our findings confirm the effectiveness of acoustic force for detritus removal in seed sorting, offering several advantages over traditional methods. The system accurately ejects seeds, with stones largely unaffected and sticks filtered due to their lower sphericity. The system handles dense groupings of similarly sized materials along the Y-axis with reasonable accuracy, and multiple passes further improve sorting precision. Performance can be further enhanced by optimizing the separator's position and height, reducing belt friction, and refining the detection software for more precise targeting of seeds. Improved cameras could also enhance tracking and enable mid-air ejection, boosting overall system efficiency and improving achievable purity and yield. Finally as noted before, reducing the variability in system latency would allow for more accurate focal point positions, leading to more consistent ejections distances.

### Color sorting
In a final experiment, we constructed a system capable of sorting seeds by color into three categories based on varying ejection forces. This is an approach not typically achievable with binary pneumatic air jet actuators. To demonstrate the system's ability to modulate ejection force, we tested its effectiveness at sorting 60 bell pepper seeds, categorized into three color groups: 20 unpainted (plain), 20 painted green, and 20 painted red. We positioned three bins beneath the conveyor belt to collect the seeds according to their categories. Plain seeds were not ejected and followed their natural trajectories into the first bin. Red seeds were ejected with medium force into the central bin, while green seeds were ejected with maximum force into the furthest bin (see Fig. 6). The weight of the paint contributed only slightly to the seeds' mass (plain: 5.9 mg; red paint: 6.3 mg; green paint: 6.2 mg, mean mass over the 20 seeds in each category).

Using the data from Fig. 4, target pressure values of 0 Pa, 5.3 kPa, and 7.0 kPa were selected. Focal points were generated with an ejection time of 15 ms, 4 mm in front of each seed when they were 8 mm from the end of the conveyor's flat section. The vertical distance from the conveyor surface to the table was 23.5 cm. Three bins with interior dimensions of 7.4 cm by 14.4 cm were positioned on the table at

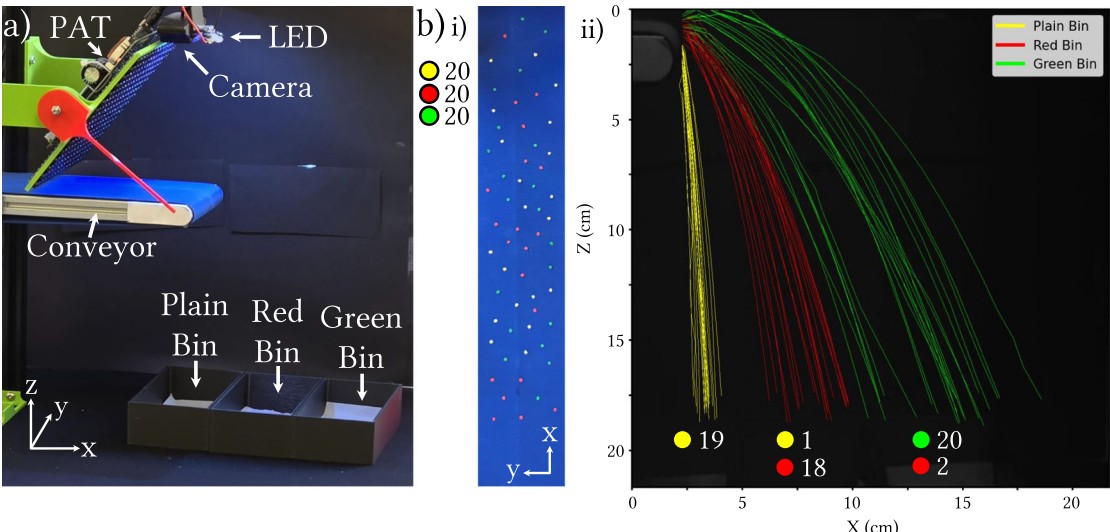

**Fig. 6 | Colored seed sorting experiments. a** Experiment setup. Seeds are hit at the end of the conveyor with acoustic radiation force from the phased array transducer board, with the force applied depending on the color detected by the camera. Plain seeds are ignored, with red seeds being ejected into the middle bin, and green seeds into the furthest bin from the belt. **b** (i) In the experiment, we have 60 seeds total, 20 of each color in the layout shown. (ii) The resulting trajectories, showing a success rate overall of 95%, with output purities for plain, red, and green seeds being 100%, 94.7%, and 90.9% respectively, and corresponding yields of 95%, 90%, and 100%.

horizontal distances of  −6.5 cm, 2.5 cm, and 10.5 cm from the conveyor's outlet edge (in the $X$-axis). The belt speed was set to its lowest at approximately 1.5 cm/s.

Using the definitions in Eqs. (1) and (2), we assess sorting performance as follows: output purities for plain, red, and green seeds were 100%, 94.7%, and 90.9%, respectively, with corresponding yields of 95%, 90%, and 100%. One plain seed ended up in the red bin, and two red seeds were found in the green bin. As detailed in Section "Static ejection experiments", while the system effectively handles tightly packed seeds along the $Y$-axis, closely spaced seeds along the $X$-axis can be ejected simultaneously due to the elongated shape of the focal point and the angle of the phased array relative to the conveyor. This likely explains the misclassification of one plain seed and two red seeds. Additionally, five green seeds were ejected beyond their designated bin; however, we included these in the bin count, as a fully realized system would be able to capture them correctly with appropriately positioned chutes.

The throughput $T$ of our system can be measured in objects ejected per second as:

$$T = (1/t_e) \cdot N_f \cdot N_b \qquad (3)$$

Where, $t_e$ is time required to eject a single seed, $N_f$ is the number of focal points an individual board is able to use simultaneously for ejection and $N_b$ is the number of boards making up the ejection system in its entirety. We currently only use a single board with a single focal point at a time, and so, given the ejection time of 15 ms, the expected maximum throughput of the system is $T = 1/(15 \times 10^{-3}) = 66$ seeds per second. There are multiple ways to improve this number, however. In addition to the obvious solutions of decreasing $t_e$ for smaller, lighter objects or increasing $N_f$ or $N_b$ (which would proportionally increase the cost and energy consumption of the system), another approach could involve ejecting multiple seeds simultaneously from a single focal point. Although, as with increasing $N_f$, this method would be limited by the total power of the board available for generating focal points. For instance, while we cannot sort two seeds into the green bin simultaneously since ejecting them requires 100% of the board's power we can eject two seeds into the red bin at once, as each only requires 50% of the board's power.

## Discussion

Our experiments validate the potential of the ultrasonic PAT system as a versatile, efficient, and precise solution for seed sorting and manipulation.

The static ejection experiments highlighted the system's ability to selectively eject seeds with a minimum horizontal spacing of 7 mm and vertical spacing of 5 mm. These values are consistent with the full-width at half-maximum (FWHM) of the focal point, demonstrating excellent spatial resolution. The ability to focus acoustic pressure at specific points without influencing neighboring seeds underscores the precision of the system. This level of control is particularly valuable for densely packed sorting applications where traditional methods struggle with overlapping or adjacent objects. During on-belt ejection, the PAT system achieved precise ejection control through modulation of pulse duration and pressure amplitude. In the detritus extraction experiments, lighter objects, such as seeds and sticks, responded more readily to changes in acoustic force, while heavier items, like stones, exhibited minimal displacement. This differentiation highlights the system's capability to sort objects based on weight and responsiveness to acoustic forces, making it suitable for a range of agricultural and industrial sorting applications.

The detritus extraction experiments demonstrated sorting accuracies of up to 73% yield, even under stress-test conditions with exaggerated detritus ratios. While slightly lower accuracy rates of 68% yield were observed with higher throughput and denser concentrations of detritus, the results remained promising given the unrealistic test conditions. Importantly, these results validate the system's ability to separate seeds from detritus with low rates of contamination, leveraging sphericity as an effective sorting metric. The findings suggest that, under real-world conditions with lower detritus levels, the system's performance would likely exceed these benchmarks. The color sorting experiments underscored the system's versatility for non-binary sorting. Achieving an overall accuracy of 95%, the PAT system demonstrated the ability to sort seeds into multiple categories with precise modulation of ejection forces. This functionality represents a significant advancement over traditional pneumatic systems, which are typically limited to binary separation.

Compared to conventional mechanical, pneumatic, and optical sorting methods, the PAT system offers several distinct advantages.

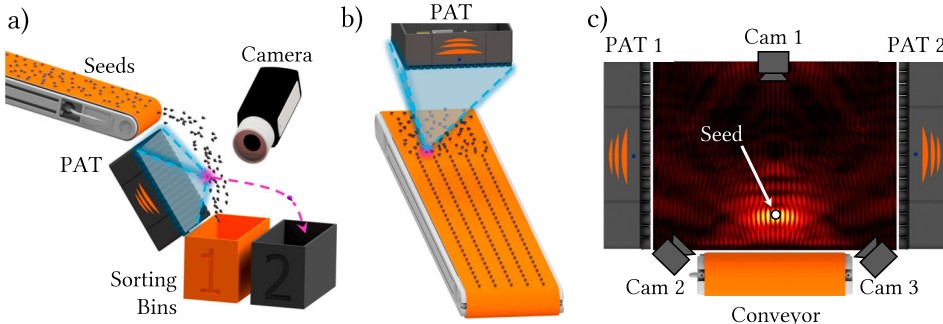

**Fig. 7 | Future applications. a** Free-fall ejection: as seeds exit the conveyor, they are scanned by a camera and ejected in mid-air. This setup requires precise real-time alignment between the camera and ejection system, but also offers key advantages: reducing the risk of ejecting desirable items along with undesirables and minimizing jamming and collisions. **b** Object pre-arrangement: a focal point sweeps the surface of the conveyor belt pushing or hovering randomly distributed seeds into a grid pattern before a later ejection step. **c** Acoustophoretic 3D inspection, coating, pelleting and priming: acoustophoresis can streamline seed processing by levitating seeds in mid-air for 360-degree inspection, coating, pelleting, and priming without physical contact, ensuring uniform treatment.

Mechanical systems often suffer from wear and tear, requiring frequent maintenance, and can cause physical damage to fragile seeds. Pneumatic systems, while fast, are noisy, energy-intensive, and prone to ejecting multiple objects simultaneously, reducing precision. Additionally, pneumatic systems can become clogged by dust and debris, which obstructs the airflow and requires regular cleaning. In contrast, our PAT-based acoustophoretic sorting system operates with low energy consumption (0.4 kW/m of conveyor belt), low noise (approximately 55 dB), and minimal physical wear, offering a cost-effective, scalable alternative. Finally, the piezoelectric transducers in our system have no moving parts, significantly reducing the risk of clogging and maintenance requirements because there are no components that can wear out or accumulate dust. For a more in-depth comparison between traditional sorting approaches and our acoustophoretic system, see Supplementary Note 1.

While our sorting system demonstrates many advantages over state-of-the-art alternatives, several limitations emerged during testing. The system's performance was influenced by object spacing and belt friction, with densely packed objects occasionally causing unintended ejections. This was especially common with the lightweight sticks in the detritus extraction experiment, which were ejected alongside neighboring seeds or stones. Our experiments show that while the PAT system's precision is well-suited for controlled environments, further refinements are needed to adapt it for high-throughput, industrial-scale operations. For example, thorough testing with an industry-standard vision system, detailed latency analysis, and calibration of parameters such as $t_e$, $N_f$, and $N_b$ to match grain size, weight, shape, and throughput requirements of the specific industry system would be essential. Addressing these limitations will enhance its robustness and scalability.

The versatility of the PAT system extends beyond the experiments conducted, with several potential applications that could revolutionize seed processing and sorting (Fig. 7).

A promising future direction involves implementing free-fall ejection, where seeds are sorted as they fall through the air. Seeds can enter free-fall either from the surface of a conveyor or after being dropped down a chute. These different presentation modes differ from setup to setup and may result in varying seed speeds, trajectories, and different positioning requirements for the acoustophoretic separation system. A calibration step would be required to find the optimal position in each case. Free-fall sorting approach eliminates surface friction and mechanical constraints, relying solely on their natural trajectories under gravity and the applied acoustic forces to direct objects into designated collection bins. It also increases the window in which ejection can occur from a line along the outlet of the conveyor belt to a 3D volume of space beneath the conveyor

(see Fig. 7a). Free-fall ejection could improve sorting accuracy by minimizing the risk of ejecting desirable seeds alongside detritus (or vice-versa) and reducing the likelihood of collisions or jamming. While this configuration requires precise real-time alignment between detection and ejection systems, advancements in low-latency cameras and acoustic control algorithms could make this a practical and efficient solution.

Another potential innovation involves using acoustic traps to pre-arrange seeds on the conveyor belt (Fig. 7b). By creating grid-like patterns or lanes, the PAT system could optimize spacing and orientation before sorting. This pre-arrangement would enhance overall sorting efficiency, particularly for dense or irregular seed distributions. Dynamic acoustic corridors and rotational manipulation could further improve object alignment, enabling seamless integration with existing sorting pipelines. Some initial results for this application can be seen in Supplementary Note 5 and Supplementary Movie 2.

The PAT system's ability to levitate and manipulate objects in three dimensions (Fig. 7c) opens opportunities for advanced seed processing. Applications such as 360-degree inspection, uniform coating, pelleting, and priming can be performed without physical contact, reducing damage to delicate seeds. By combining these steps into a single levitation-based process, the system could streamline seed preparation, enhancing consistency and quality while reducing operational complexity. Lifting of a single seed is shown in Supplementary Note 6 and Supplementary Movie 3.

Beyond agriculture, our technology could be extended to related industries, such as pharmaceuticals, where it could handle fragile capsules or tablets with similar precision. The PAT system's non-contact capabilities make it suitable for delicate object handling in industries such as electronics recycling and food processing. By minimizing physical stress and ensuring precise manipulation, this technology offers a sustainable solution for sorting and processing fragile or small components.

In summary, our ultrasonic phased array transducer system represents a substantial advancement in seed sorting, providing a contactless, efficient, and accurate method for handling delicate materials like seeds. This technology offers several advantages over traditional mechanical and pneumatic sorting methods, including high-speed operation, low noise levels (approximately 55 dB), energy efficiency (0.4 kW/m of belt), and cost-effectiveness. The experiments we have detailed validate the system's effectiveness, and the applications we showcase of detritus extraction and color sorting, demonstrate its practical applicability in real-world scenarios.

This technology could significantly benefit agriculture by improving sorting efficiency and cost-effectiveness. Ensuring that only high-quality seeds are planted will boost crop yields and reduce waste.

Furthermore, its advanced capabilities for hovering and 3D manipulation could streamline seed treatment and quality assurance, ultimately improving seed quality and performance.

Our ultrasonic PAT system offers a promising foundation for transforming seed separation and related industries. Commercial sorters for high-value seeds routinely achieve purities and yields above 99%, with some applications requiring 99.9% purity[39]. These results are achieved under controlled conditions using highly optimized systems and pre-cleaned inputs (typically 5–15% detritus). In contrast, our acoustophoretic separation system is a proof of concept designed to demonstrate contactless ejection via ultrasonic phased arrays. To test its core capabilities, we deliberately introduced detritus levels far exceeding realistic scenarios, achieving 70–100% purity and yield depending on application and sorting passes. While these results are encouraging, further development is needed to meet commercial standards. Future work could explore advanced algorithms for object pre-arrangement and free-fall ejection to improve separation performance. Additionally, integration with high-performance optical systems would also give a fuller picture of a complete sorting system. Larger-scale experiments with realistic seed-to-detritus ratios, higher throughput, and additional trials would help assess performance under practical conditions and provide statistically robust evaluations of trajectory variability. This, in turn, would support finer acoustic tuning for material-specific sorting. With continued research, the ultrasonic PAT platform holds strong potential to drive more efficient, sustainable, and advanced agricultural technologies.

## Methods

Our acoustophoretic sorting system was designed for high-speed, non-contact seed ejection. Our system utilizes ultrasonic PATs to create and control high-frequency sound waves for seed ejection. The system precisely focuses sound waves to eject objects by deflecting them from their natural paths, offering an alternative to traditional mechanical and pneumatic sorting methods.

### Phased array transducers

We used the Hardware Evaluation Kit from AcoustoFab Ltd, which is a grid arrangement of 16 × 16 transducers. Each transducer (HongChang Electronic, HC10T-40TR-P) in this array has a diameter of 10 mm and operates at a consistent frequency of 40 kHz, delivering 5.46 Pa at a distance of 1 m when driven at 20 volts peak-to-peak. An FPGA (field-programmable gate array) board is mounted on the array to control the phase and amplitude of each transducer at high update rates up to 40,000 updates per second. As shown in Fig. 1, our PAT board is fixed to a conveyor belt (AS Conveyor Systems) and tilted 45 degrees, facing the edge of the belt.

Ejecting an object, such as a seed, from its path can be achieved by focusing acoustic waves on the object's side. To create a focal point at position $\mathbf{x_p}$ with a target phase $\phi'_p$ (typically set to zero for single point focusing), the phase of each transducer $\phi_t$ can be defined as follows:

$$\phi_t = -k|\mathbf{x_p} - \mathbf{x_t}| + \phi'_p, \qquad (4)$$

where $k$ represents the wave number for the frequency used and $\mathbf{x_t}$ represents the position of each transducer. Figure 2a shows simulated acoustic pressure amplitudes when creating a focal point at different positions. These simulations were generated using the transducer piston model[25,40], a well established theoretical framework that describes how piezoelectric transducers convert electrical energy into mechanical vibrations to generate acoustic waves. When the focused sound wave interacts with an object, an acoustic radiation force $\mathbf{F_a}$ is generated, pushing the object out of its trajectory. Note that the strength and direction of this force depend on the pressure distribution along the object's surface.

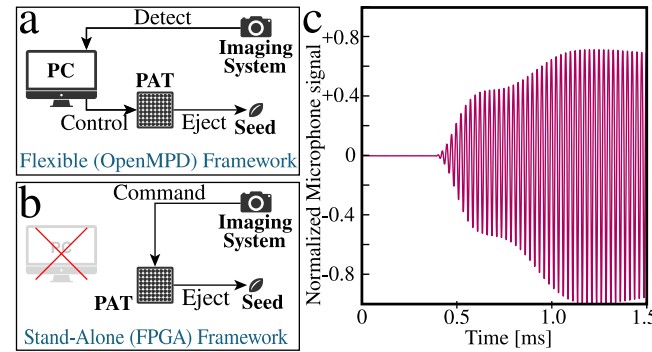

**Fig. 8 | Two types of acoustophoretic ejection framework. a** Flexible framework based on the OpenMPD software platform. **b** Stand-alone framework utilizing the mounted FPGA for phase and amplitude computation. **c** Plot of the normalized signal latency from the microphone. Source data for this figure are provided in Supplementary Data 3.

### Acoustophoretic ejection frameworks

In this study, we proposed two acoustophoretic ejection frameworks (Fig. 8a, b): a flexible framework based on the OpenMPD software platform[41] and a stand-alone framework utilizing the mounted FPGA for phase and amplitude computation similar to ref. 24.

In the flexible framework, a PC processes image data captured by a camera (LogiTech, C920 HD Pro Webcam) mainly to detect seed positions and then controls the PAT to create a focal point to eject the detected seeds (see Fig. 8a). Our software is based on the OpenMPD platform[41], which facilitates the creation of focal points with high spatial-temporal accuracy. Given the position, the pulse duration, and the target pressure amplitude of the focal point, our software optimizes the phases and amplitudes of the transducers using the GS-PAT algorithm[25] and sends them to the PAT at high rates (more than 10,000 updates per second). While this paper focuses on ejecting a single seed at a time, the OpenMPD framework also supports the creation of multiple focal points at the same time[41].

The seed detector integrated into our software is based on the OpenCV library and can detect seed positions in real-time after simply executing background subtraction, image thresholding, and erosion and dilation. In addition to seed positions, the detector can provide other information about seeds, such as size, sphericity, and color, and this additional information can be used as sorting criteria.

The reason why we call this framework flexible is that we can easily build and customize our own applications thanks to the two existing OpenMPD and OpenCV platforms for both sound control and image processing. To accelerate the development and evaluation process, we are using this framework for most of the experiments and applications presented in this paper.

In the stand-alone framework, phase optimization (Eq. (4)) is implemented using the FPGA, which is already mounted on the PAT to control the transducers. Thus, the PAT can eject seeds by receiving simple commands (e.g., Y-coordinate) from the imaging system, without the need for the PC and GPU as intermediaries (see Fig. 8b). This does not only reduce the cost and energy consumption of the system but also enhances its modularity (i.e., makes integration into existing sorting platforms easy). The ultimate goal is to replace pneumatic ejectors used in optical sorters without significant changes in imaging systems or other parts.

Another advantage of this stand-alone framework is its low latency. To measure the latency of the framework, we used a microphone (Brüel & Kjær 4138-A-015) connected to an oscilloscope (PicoScope 4262). A microcontroller sent a command to the FPGA on the PAT and at the same time sent a trigger signal to the oscilloscope. Immediately after receiving the command, the FPGA instructed the

PAT to create a focal point at a distance of 12 cm from the PAT center point, while the oscilloscope began sampling the sound pressure values at the focal point in response to the trigger signal. Figure 8c shows normalized signal from the microphone. The sound pressure reaches 95% of its steady state 1.08 ms after receiving the trigger signal, thus, the latency of this framework is about 1 ms, which is faster than a typical pneumatic jet system with a latency of ~ 10–30 ms [42,43].

## Data availability
All data generated in this study are provided in the Source Data file.

## Code availability
The code used in this paper is available from the corresponding author upon request. All code is based on the OpenMPD platform available at https://github.com/RMResearch/OpenMPD.

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

## Acknowledgements

Support for this work was provided by the Royal Academy of Engineering through its Chairs in Emerging Technology Programme (CIET 17/18) for J.H. and Z.M., and the UKRI Frontier Research Guarantee grant (EP/X019519/1) for R.H. The authors would like to thank Yihan Dong from Acoustofab Ltd for her support in creating videos and figures.

## Author contributions

J.H. and R.H. conceived the project. R.H. supervised the project. J.H., Z.M. and R.H. designed, executed, and analyzed the experiments. All authors also developed the discussion and were responsible for writing and revising the manuscript. At Acoustofab Ltd, we initiated the project based on customer engagement, developed the acoustic separation system, and conducted the experiments to acquire data. University College London provided additional academic support, including leading the manuscript preparation, supplying tools for data visualization, and contributing to feedback and discussion.

## Competing interests

J.H., Z.M., and R.H. are employed by Acoustofab Ltd, which partially funded this work. R.H. also holds shares in Acoustofab Ltd. All authors are listed as inventors on United Kingdom Patent No. 2503660.9 which is based on the work described in this article.
