## [Transparent Peer Review file · Nature Communications]

Acoustophoretic System for Seed Separation on Conveyor Belts

Corresponding Author: Dr James Hardwick

Version 0:

Reviewer comments:

Reviewer #1

(Remarks to the Author)

In this paper, the authors intend to present a novel sorting approach using ultrasonic phased array transducers. First of all, the presented topic is of great interest in the context of the development bulk good sorting systems (especially for small objects) and has also a high scientific impact. In principle, the proposed solution can be used to sort “not cooperative” materials. In conclusion, the topic of the paper is of big relevance.

To improve the paper, I have the following comments:

1. The authors should check a more precise wording in the context of the state of the art:
 - a. For specialists in the sensor-based sorting domain, the topic the authors present is not a novel sorting approach, but a novel separation approach.
 - b. Regarding the price of the sorters, you should compare only the costs of the different separation methods, not the price of the sorting system itself. The mentioned prices are mainly depending on the sensor technology.
 - c. Regarding the state of the art, the sorters are often equipped with hyperspectral or multispectral cameras in SWIR or NIR, not only color in the RGB domain (page 1, line 46).
 - d. Page 1 line 77/78: please more details what exactly is not fulfilled by the other technologies
 - e. Page 2 line 143: what is a “notable” accuracy?
2. Missing is a more detailed description of the state of the art in the context of the used methods. The authors compare their new solution only with other separation techniques. Ongoing work, national and international, in the field of ultrasonic separation solutions, should be mentioned and judged. If there is no work existing, then this should also be stated in the paper.
3. The wording “free fall ejection” is correct, but based on the material presentation approach we can distinguish four different bulk good presentation modes: free fall (90 degree), belt sorting (your free fall ejection), chute sorting and channel sorting. All these methods result in different trajectories of the material and thus also on the requirements of the separation technology.
4. The evaluation of the system is presented in a good way. The design of the experiments are described in a good detail detail. Can you please state what the impact of e.g. dust in the material composition influences the separation results?

Further comments:

Page 3 line 204: missing space after “Fig.”

Page 6 line 434/435: the results are based on my experience not excellent. Can you say a few words about the needed accuracy based on the application domain?

Page 8 line 540: the improvement of the result is possible by using more boards. How does this effect the costs of the system, and the energy consumption?

Page 9 line 623: what are the requirements in the industrial domain?

Page 10 line 706: very general, can you state more detailed your ideas about using machine learning in your domain?

Page 10 line 708: combination of acoustic sorting with optical sorting? How?

Line 975: Belt width: it is not obvious why the gravity and air jet sorting machines are not flexible regarding the flexibility of

band width?

(Remarks on code availability)

Reviewer #2

(Remarks to the Author)

Key results

This study presents an approach based on ultrasonic phased array transducers to physically separate several types of seeds and other types of small objects that are considered contaminants. The spatial resolution of the separation technique is investigated, along with its ability to sort a specific type of seed from sticks and stones, which represent detritus in the mix.

Validity

The authors claim that the superior control of the system makes the approach valuable for industrial sorting applications with overlapping or adjacent objects on densely packed conveyor belts. This claim seems to be conflicting with the indication of the authors that a minimum horizontal spacing of 7 mm and a minimum vertical spacing of 5 mm are necessary for the seeds to be selectively ejected. These conditions can hardly be considered as a dense packing, especially when compared to gravity sorting setups, like the one presented in the authors' video, where the packing density and the throughput are significantly higher. In terms of packing density, the indicated limitations of the proposed solution are similar to those of pneumatic ejection systems. As an alternative for those type of sorting techniques, the proposed approach is definitely interesting. When comparing the proposed technique to gravity sorters, a more thorough discussion on the throughput and sorting performance of both techniques could be useful. The authors have performed experiments that are rather limited in scale to assess the performance of the sorting approach. The performance in those experiments is expressed in terms of accuracy, while yield and purity are more common performance metrics in sorting applications. Usually, there is a required output purity that has to be attained after sorting to consider the sorting successful. For the application of seed sorting, the authors explain that removing detritus is valuable but mention no concrete target in terms of detritus removal that should be achieved. The level of detritus in the input is also deliberately different than usual. All these factors make it hard to evaluate whether the technique has a suitable sorting performance for the discussed application. The difference in throughputs between the compared sorting techniques is also described only vaguely. It remains therefore unclear whether the possible higher sorting performance of the proposed approach due to its higher ejection selectivity can justify the probably much lower throughput compared to gravity sorters. Because of the limited scale of the experiments and the lack of evidence that the approach can achieve acceptable levels of detritus after sorting a typical input in a limited number of runs, perhaps the conclusions should be formulated somewhat more modestly. I consider the technique an interesting alternative to pneumatic sorting, that probably has sorting performances and throughputs in the same order of magnitude. The experimental back-up for a thorough comparison with gravity sorters is still missing.

Significance

The proposal to use ultrasonic phased array transducers to sort small objects is original and significant, as it offers an alternative to existing separation techniques that each have their own limitations. While this paper does not offer representative experimental proof for industrial-scale sorting with the proposed technique, it clearly demonstrates the ability to selectively eject small objects with reasonable spatial resolution. As such, an interesting proof of concept is delivered with this paper. The suggestions of the authors for future research are also valid and interesting.

Data and methodology

In the experiments related to Figure 4, each trial was conducted five times per object and per pulse duration or pressure amplitude. This number of trials seems sufficient to demonstrate that the displacement of the objects is proportional to both the pulse duration and pressure amplitude, up to some point of saturation, where increasing these values no longer leads to an increased displacement. However, there is often a significant spread in the displacement values among the five trials under the same conditions. The ranges of displacement values of the different objects strongly overlap. To get an understanding of which objects have a statistically significant difference in displacement over the chosen pulse duration and pressure amplitude ranges, a much higher number of trials would have to be conducted.

In the experiment of the detritus removal, the authors chose a seeds to stones and sticks ratio that allowed to demonstrate a significant reduction of the number of sticks and stones after two runs with a relatively limited number of objects in the input mix. This choice is understandable, as it leads to an insight into the sorting performance with manageable experimental work. However, a larger experiment with a representative input mix and clear targets in terms of maximum detritus levels in the output would be even more interesting (perhaps in future work) to evaluate if this target can be achieved by the proposed method in an acceptable number of runs.

Analytical approach

The analytical work behind the ultrasonic phased array transducer system and the focusing algorithm presented in the appendix is impressive. The interpretation of the results of the experiments is rather simple, since the limited number of ejected objects does not allow for more advanced statistical analysis.

Suggested improvements

As mentioned before, a larger number of trials would allow to make stronger conclusions on the separation performance of the approach. The experiments would also be more directly relevant for industrial applications if the experiments would be conducted with a representative input mix and if more common metrics would be used to express the performance. The value of the approach with respect to competing sorting techniques can only be made clear if throughput, purity and yield are discussed starting from a representative mix. The aspects included in the comparison with traditional approaches that is presented in the appendix are not sufficient to have a convincing comparison. If this is not considered within the scope of the manuscript, at least this should be more clearly proposed as an idea for future work.

Clarity and context

The clarity and accessibility of the text are excellent. The quality of the figures, tables and videos is outstanding, which helps to better understand the results. The context is quite clear, except that the targets when sorting seeds and the typical inputs in this sector are not really discussed. It is only clear that lower quality seeds and detritus are undesired and therefore removed during sorting. Some quantitative information could help to judge whether the reported sorting accuracies are close to what is required for this application or not.

References

The quantity and quality of references seems appropriate and to the best of my knowledge, no critical references seem to be forgotten.

(Remarks on code availability)

Version 1:

Reviewer comments:

Reviewer #1

(Remarks to the Author)

Dear authors,

thank you very much for your changes in the document. I have no further comments.

Best regards

Reviewer 1

(Remarks on code availability)

Reviewer #2

(Remarks to the Author)

Thank you for the clarifications and revision of the manuscript. My concerns have been adequately addressed by the authors. Therefore, I deem the manuscript suitable for publication.

(Remarks on code availability)

We would like to sincerely thank both reviewers for their thoughtful, detailed, and constructive feedback on our manuscript. We are grateful for the time and effort invested in evaluating our work and for the kind words regarding the paper's originality, clarity, and potential significance. Your comments have helped us substantially improve the quality and clarity of the manuscript.

Please find below our detailed responses to each point raised. Changes made in response to the reviewers' suggestions are highlighted in red in the revised manuscript to facilitate your review.

Reviewer 1:

In this paper, the authors intend to present a novel sorting approach using ultrasonic phased array transducers. First of all, the presented topic is of great interest in the context of the development bulk good sorting systems (especially for small objects) and has also a high scientific impact. In principle, the proposed solution can be used to sort "not cooperative" materials. In conclusion, the topic of the paper is of big relevance.

To improve the paper, I have the following comments:

1. The authors should check a more precise wording in the context of the state of the art:
 - a. For specialists in the sensor-based sorting domain, the topic the authors present is not a novel sorting approach, but a novel separation approach.
 - b. Regarding the price of the sorters, you should compare only the costs of the different separation methods, not the price of the sorting system itself. The mentioned prices are mainly depending on the sensor technology.
 - c. Regarding the state of the art, the sorters are often equipped with hyperspectral or multispectral cameras in SWIR or NIR, not only color in the RGB domain (page 1, line 46).
 - d. Page 1 line 77/78: please more details what exactly is not fulfilled by the other technologies
 - e. Page 2 line 143: what is a "notable" accuracy?

A: We have made changes in the updated manuscript to address these reviewer concerns:

- a. We agree with the reviewer that the distinction between "sorting" systems and their vision and separation components was unclear. We have replaced "acoustic sorting" with "acoustic separation" throughout the manuscript where appropriate, particularly in the introduction, where we have restructured and updated the discussion of gravity sorting,

optical sorting, their component vision and separation systems, and our acoustic separation contribution.

- b. We have updated the introduction and Section 1 of the SI so that the price comparisons between separator formats are now fair. We compare our acoustic ejection system against the separation component of optical sorters - the array of nozzles and air compressor - excluding the cost of the vision system, which makes up the bulk of the cost for an optical sorting system as correctly pointed out by the reviewer. In contrast, gravity sorters are compared as entire units as they do not include such vision systems.
 - c. We have included a discussion of the different vision formats an optical sorter may incorporate when we introduce them in paragraph 3 of Section I.
 - d. We have removed this line in the revised manuscript as we agree with the reviewer that it was confusing as previously written. The manuscript now simply highlights challenges of robotic approaches.
 - e. We have replaced “notable” with a number representing the achieved accuracy, yield and purity of our system (i.e., 60–100%).
2. Missing is a more detailed description of the state of the art in the context of the used methods. The authors compare their new solution only with other separation techniques. Ongoing work, national and international, in the field of ultrasonic separation solutions, should be mentioned and judged. If there is no work existing, then this should also be stated in the paper.

A: While there has been a lot of prior work using phased arrays of transducers for acoustic levitation, which we discuss in the paper, to our knowledge there has been no prior work using them to push millimetre-scale objects, certainly not for sorting purposes. We have updated paragraph 6 in the introduction to reflect this and have added an additional short paragraph directly after to discuss prior work discussing ultrasound-based actuation in other application domains (i.e., acoustofluidics and human-computer interaction).

3. The wording “free fall ejection” is correct, but based on the material presentation approach we can distinguish four different bulk good presentation modes: free fall (90 degree), belt sorting (your free fall ejection), chute sorting and channel sorting. All these methods result in different trajectories of the material and thus also on the requirements of the separation technology.

A: We agree that the discussion of seed presentation modes was a little confusing as previously written in the Free-fall ejection section of the discussion. We have added a few sentences to the updated manuscript to clarify that our system could work with any of these modes (from outlets of either belts or chutes and at a variety trajectory angles) but would require an optimisation of position for

the acoustic separation system in each case.

4. The evaluation of the system is presented in a good way. The design of the experiments are described in good detail. Can you please state what the impact of e.g. dust in the material composition influences the separation results?

A: Dust and particulates are known to pose a significant challenge for air-jet separation systems, as nozzles often clog, requiring frequent cleaning and maintenance, which increases costs and reduces uptime. In contrast, our system uses piezoelectric transducers that generate sound waves through the air, avoiding moving parts or porous surfaces where dust could accumulate, thus minimizing clogging risk. Additionally, the transducers' high-frequency vibrations can help dislodge or prevent particle buildup, further reducing clogging. We have added a sentence in the updated manuscript on this topic to the paragraph in the discussion section comparing our PATs and air-jet separators directly (section III, paragraph 4).

Further comments:

- Page 3 line 204: missing space after “Fig.”

A: We have fixed this error.

- Page 6 line 434/435: the results are based on my experience not excellent. Can you say a few words about the needed accuracy based on the application domain?

A: We have updated our metrics from accuracy to purity and yield to better align with industry standards for sorting. This should provide clearer insights into our results. Additionally, we revised the final paragraph of the discussion to address performance requirements in the sorting industry, positioning our device as a promising future technology rather than a fully developed product ready for direct comparison with industry-standard machines.

- Page 8 line 540: the improvement of the result is possible by using more boards. How does this effect the costs of the system, and the energy consumption?

A: The cost and energy consumption would of course increase proportionally with each additional board added to tile the width of a belt. However, as this is also true for air-jet separators, we believe that our claim of lower running cost and energy consumption vs compressed air per unit width of the conveyor holds true. We have added a sentence stating that the potential solution of using more boards would proportionally increase the cost and energy consumption of the system (Colour Sorting Section – 4th Paragraph).

- Page 9 line 623: what are the requirements in the industrial domain?

A: For example, thorough testing with an industry-standard vision system, detailed latency analysis, and calibration of parameters, such as t_e (time required to eject a single seed), N_f (number of focal points per board), and N_b (number of boards), to match grain size, weight, shape, and throughput

requirements of the specific industry system would be essential. We have included this sentence at this location in the updated document to clarify.

- Page 10 line 706: very general, can you state more detailed your ideas about using machine learning in your domain?

A: We have removed this claim in the new final paragraph of the discussion section in the updated manuscript as we agree that it was as an overly general and unfounded claim.

- Page 10 line 708: combination of acoustic sorting with optical sorting? How?

A: We meant a combination of our acoustic separation system and detection sensors used in optical sorters. We have removed this sentence in the updated manuscript to eliminate potential confusion.

- Line 975: Belt width: it is not obvious why the gravity and air jet sorting machines are not flexible regarding the flexibility of band width?

A: We agree with the reviewer and have removed a large part of the Belt Width and Flexibility subsection of the 1st section in the SI. Gravity sorting plates and air jet separators can indeed be scaled up or down to fit any desired width. We have left the section of the paragraph describing the range of acoustic sorters and the added flexibility in positioning that gives them relative to the other components of the sorting system.

Reviewer 2:

Key results

This study presents an approach based on ultrasonic phased array transducers to physically separate several types of seeds and other types of small objects that are considered contaminants. The spatial resolution of the separation technique is investigated, along with its ability to sort a specific type of seed from sticks and stones, which represent detritus in the mix.

Validity

- The authors claim that the superior control of the system makes the approach valuable for industrial sorting applications with overlapping or adjacent objects on densely packed conveyor belts. This claim seems to be conflicting with the indication of the authors that a minimum horizontal spacing of 7 mm and a minimum vertical spacing of 5 mm are necessary for the seeds to be selectively ejected. These conditions can hardly be considered as a dense packing, especially when compared to gravity sorting setups, like the one presented in the authors' video, where the packing density and the throughput are significantly higher. In terms of packing density, the indicated limitations of the proposed solution are similar to those of pneumatic ejection systems. As an alternative for those type of sorting techniques, the proposed approach is

definitely interesting. When comparing the proposed technique to gravity sorters, a more thorough discussion on the throughput and sorting performance of both techniques could be useful.

A: We agree that the term "dense packing" warrants further clarification in the context of our system's capabilities.

In the final paragraph of Static Ejection Experiments Section of the updated manuscript, we have added an additional discussion to clarify that while our system achieves selective ejection with minimum spacing of 7 mm (horizontal axis) and 5 mm (vertical axis), we now explicitly acknowledge in the discussion that these numbers are only relevant in comparison to air-jet systems, which eject objects from a two-dimensional monolayer and are not appropriate for comparison with gravity sorters whose populations of sortable objects can be several layers deep.

- The authors have performed experiments that are rather limited in scale to assess the performance of the sorting approach. The performance in those experiments is expressed in terms of accuracy, while yield and purity are more common performance metrics in sorting applications. Usually, there is a required output purity that has to be attained after sorting to consider the sorting successful.

A: We have revised the Results section of the manuscript to include analyses of both yield and purity for experiments where these metrics are appropriate and informative.

Specifically, for the detritus extraction experiments, we now report yield (the proportion of desirable seeds correctly retained) and purity (the proportion of desirable items in the accepted output) alongside the original accuracy figures. These metrics offer a more granular understanding of performance, particularly under stress-test conditions involving high detritus-to-seed ratios.

Similarly, in the color sorting experiments, we computed yield and purity for each color class, thereby quantifying the effectiveness of our non-binary sorting implementation. The updated section now includes per-class metrics and a comparative analysis that highlights the precision and selectivity of our phased array acoustic system.

- For the application of seed sorting, the authors explain that removing detritus is valuable but mention no concrete target in terms of detritus removal that should be achieved. The level of detritus in the input is also deliberately different than usual. All these factors make it hard to evaluate whether the technique has a suitable sorting performance for the discussed application.

A: We appreciate the reviewer's concern regarding the absence of a specific target metric for detritus removal. We agree that this can make it more difficult to assess system performance in absolute terms. However, as seed sorting requirements can

vary depending on the crop type, use case, and industrial context, setting a single fixed threshold (e.g., for purity or yield) would be somewhat arbitrary in the context of a prototype system.

Commercial-grade optical sorters (separation, vision and all other systems combined) often aim for purity and yield levels exceeding 99%, and anything below this may be considered suboptimal for production environments. However, our system is not positioned as a direct competitor to such systems in its current form. Rather, it serves as a proof of concept for the separation component of a full optical sorting system and aims to demonstrate that acoustic ejection can perform selective sorting of lightweight seeds with promising performance, even under exaggerated and challenging conditions.

To stress-test our system's capabilities, we deliberately used unrealistically high detritus concentrations (e.g., ~30% purity) in our input mixtures—substantially more severe than the 5–15% typical in industrial settings. Despite this, our system achieved purity and yield values in the range of 70–100%, which we believe underscores its potential as a foundation for future high-performance acoustic separation platforms. Our goal is not to claim industrial readiness at this stage, but to demonstrate that acoustic forces—when paired with a scalable architecture—could eventually reach commercial-grade performance, particularly when integrated with a high-resolution optical detection system.

- The difference in throughputs between the compared sorting techniques is also described only vaguely. It remains therefore unclear whether the possible higher sorting performance of the proposed approach due to its higher ejection selectivity can justify the probably much lower throughput compared to gravity sorters. Because of the limited scale of the experiments and the lack of evidence that the approach can achieve acceptable levels of detritus after sorting a typical input in a limited number of runs, perhaps the conclusions should be formulated somewhat more modestly. I consider the technique an interesting alternative to pneumatic sorting, that probably has sorting performances and throughputs in the same order of magnitude. The experimental back-up for a thorough comparison with gravity sorters is still missing.

A: We agree that the claims regarding industrial applicability, particularly in comparison to gravity sorters, should be stated with greater care. We have now softened the language in several sections of the updated manuscript, specifying that our system is currently best positioned as a future alternative to pneumatic systems, and that the comparison with gravity sorters is more one of cost and efficiency than a direct like-for-like comparison. The changes to language were

made in: Section I – 2nd paragraph, Section I – 8th paragraph, Section II C – 1st paragraph, and SI Section I – 1st paragraph.

Significance

- The proposal to use ultrasonic phased array transducers to sort small objects is original and significant, as it offers an alternative to existing separation techniques that each have their own limitations. While this paper does not offer representative experimental proof for industrial-scale sorting with the proposed technique, it clearly demonstrates the ability to selectively eject small objects with reasonable spatial resolution. As such, an interesting proof of concept is delivered with this paper. The suggestions of the authors for future research are also valid and interesting.

A: We thank the reviewer for their positive assessment. We agree that this work serves as a proof of concept and appreciate the recognition of its originality and potential. We are glad the proposed future directions were found to be of interest.

Data and methodology

- In the experiments related to Figure 4, each trial was conducted five times per object and per pulse duration or pressure amplitude. This number of trials seems sufficient to demonstrate that the displacement of the objects is proportional to both the pulse duration and pressure amplitude, up to some point of saturation, where increasing these values no longer leads to an increased displacement. However, there is often a significant spread in the displacement values among the five trials under the same conditions. The ranges of displacement values of the different objects strongly overlap. To get an understanding of which objects have a statistically significant difference in displacement over the chosen pulse duration and pressure amplitude ranges, a much higher number of trials would have to be conducted.

A: We agree that while five trials per condition were sufficient to illustrate general trends in displacement behavior, a more extensive dataset would be necessary to rigorously establish statistically significant differences between object classes across the tested ranges of pulse duration and pressure amplitude. Each seed that leaves the conveyor belt—whether deflected by the acoustic separator or continuing along its natural trajectory—exhibits displacement that falls within a probability distribution. Increasing the pressure amplitude or pulse duration effectively shifts this distribution forward along the x-axis, reflecting greater average displacement. We acknowledge that with the current number of trials, the spread in these distributions and their overlap between object classes limits our ability to draw strong statistical distinctions. We have clarified this limitation in the

manuscript and noted that future work should involve a larger number of trials per condition to better characterize these probability distributions and how they evolve in response to changes in acoustic parameters (final paragraph in Section III).

- In the experiment of the detritus removal, the authors chose a seeds to stones and sticks ratio that allowed to demonstrate a significant reduction of the number of sticks and stones after two runs with a relatively limited number of objects in the input mix. This choice is understandable, as it leads to an insight into the sorting performance with manageable experimental work. However, a larger experiment with a representative input mix and clear targets in terms of maximum detritus levels in the output would be even more interesting (perhaps in future work) to evaluate if this target can be achieved by the proposed method in an acceptable number of runs.

A: We agree that larger-scale tests with realistic seed-to-detritus ratios and defined output targets would strengthen the evaluation. This has now been noted in the final paragraph in Section III as a key direction for future experiments.

Analytical approach

- The analytical work behind the ultrasonic phased array transducer system and the focusing algorithm presented in the appendix is impressive. The interpretation of the results of the experiments is rather simple, since the limited number of ejected objects does not allow for more advanced statistical analysis.

A: We thank the reviewer for their positive feedback on the analytical aspects of our system. We agree that the scope for statistical interpretation is currently limited by the number of ejected objects in each experiment. As noted in the final paragraph in Section III, we plan to scale up future experiments to enable more rigorous statistical analysis of sorting performance.

Suggested improvements

- As mentioned before, a larger number of trials would allow to make stronger conclusions on the separation performance of the approach. The experiments would also be more directly relevant for industrial applications if the experiments would be conducted with a representative input mix and if more common metrics would be used to express the performance. The value of the approach with respect to competing sorting techniques can only be made clear if throughput, purity and yield are discussed starting from a representative mix. The aspects included in the comparison with traditional approaches that is presented in the appendix are not sufficient to have a convincing comparison.

If this is not considered within the scope of the manuscript, at least this should be more clearly proposed as an idea for future work.

A: As already detailed in earlier responses, we have added yield and purity metrics to the relevant sections of the Results section to better align our performance evaluation with common practices in the sorting space. We agree that a larger number of trials and experiments using representative input mixes are important for drawing stronger conclusions about industrial relevance. While a full benchmarking study is beyond the current scope, we now explicitly propose this as a direction for future work in Section III, including plans to assess throughput and enable more robust comparisons with traditional sorting systems.

Clarity and context

- The clarity and accessibility of the text are excellent. The quality of the figures, tables and videos is outstanding, which helps to better understand the results. The context is quite clear, except that the targets when sorting seeds and the typical inputs in this sector are not really discussed. It is only clear that lower quality seeds and detritus are undesired and therefore removed during sorting. Some quantitative information could help to judge whether the reported sorting accuracies are close to what is required for this application or not.

A: We thank the reviewer for their positive comments regarding the clarity of the text and the quality of the visual materials. In response to the request for more context on sorting targets and typical input conditions in seed processing, we have expanded the Discussion section to include quantitative benchmarks from the commercial sector. This updated paragraph (end of Section III) clarifies that purity and yield levels above 99%—and up to 99.9% in some applications—are standard in high-value seed sorting, typically under controlled conditions with 5–15% detritus. We contrast this with our proof-of-concept system, which was intentionally tested under more extreme input conditions, and now clearly position our results (70–100% purity and yield) as an early demonstration of feasibility rather than commercial readiness. We believe this added context helps clarify the scope and significance of our contributions.

References

- The quantity and quality of references seems appropriate and to the best of my knowledge, no critical references seem to be forgotten.

A: We are glad that the reference list was found to be appropriate and sufficiently comprehensive.